# Blockwise Self-Supervised Learning at Scale

**Shoaib Ahmed Siddiqui**                                        *msas3@cam.ac.uk*
*University of Cambridge, UK*

**David Krueger**                                                *dsk30@cam.ac.uk*
*University of Cambridge, UK*

**Yann LeCun**                                                   *yann@fb.com*
*Meta-FAIR, NY, USA*
*New York University, NY, USA*

**Stéphane Deny**                                    *stephane.deny@aalto.fi*
*Aalto University, Espoo, Finland*

**Reviewed on OpenReview:** *https://openreview.net/forum?id=M2m618iIPk*

## Abstract

Current state-of-the-art deep networks are all powered by backpropagation. However, long backpropagation paths as found in end-to-end training are biologically implausible, as well as inefficient in terms of energy consumption. In this paper, we explore alternatives to full backpropagation in the form of blockwise learning rules, leveraging the latest developments in self-supervised learning. We show that a blockwise pretraining procedure consisting of training independently the 4 main blocks of layers of a ResNet-50 with Barlow Twins' loss function at each block performs almost as well as end-to-end backpropagation on ImageNet: a linear probe trained on top of our blockwise pretrained model obtains a top-1 classification accuracy of 70.48%, only 1.1% below the accuracy of an end-to-end pretrained network (71.57% accuracy). We perform extensive experiments to understand the impact of different components within our method and explore a variety of adaptations of self-supervised learning to the blockwise paradigm, building an exhaustive understanding of the critical avenues for scaling local learning rules to large networks, with implications ranging from hardware design to neuroscience. Code to reproduce our experiments is available at: https://github.com/shoaibahmed/blockwise_ssl.

## 1 Introduction

One of the main components behind the success of deep learning is backpropagation. It remains an open question whether comparable recognition performance can be achieved with local learning rules, a question that dates back to the early days of deep learning when Hinton et al. (2006); Bengio et al. (2006) proposed a greedy layerwise training algorithm for deep belief networks (DBN). Recent attempts in the context of supervised learning and unsupervised learning have only been successful on small datasets like MNIST (Salakhutdinov & Hinton, 2009; Löwe et al., 2019; Ahmad et al., 2020; Ernoult et al., 2022; Lee et al., 2015) or large datasets but small networks like VGG-11 (Belilovsky et al., 2019) (67.6% top-1 accuracy on ImageNet).

Being able to train models with local learning rules at scale is useful for a multitude of reasons. By locally training different parts of the network, one could optimize learning of very large networks while limiting the memory footprint during training. This approach was recently illustrated at scale in the domain of

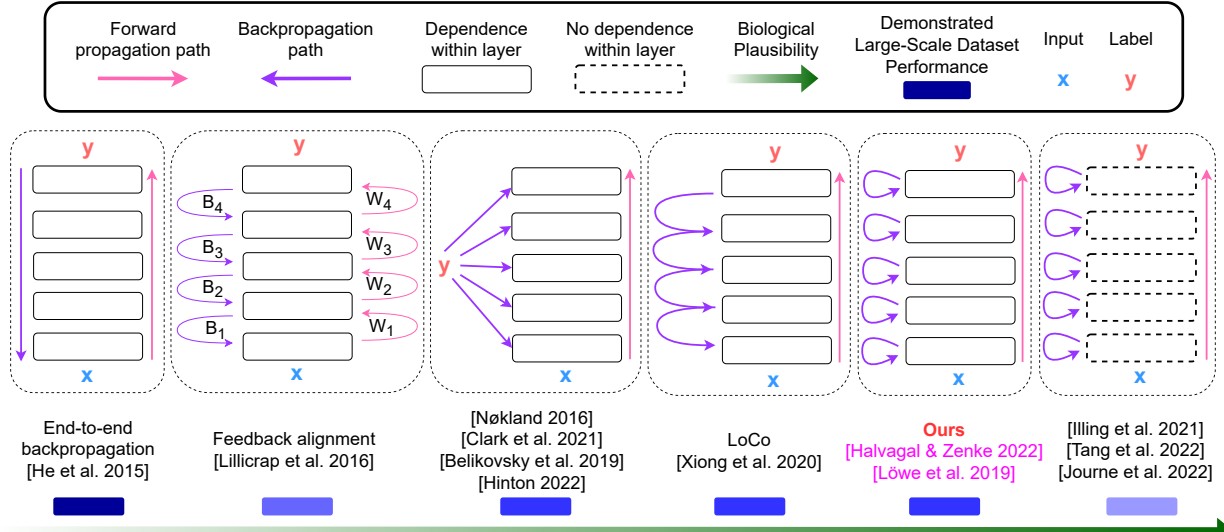

Figure 1: We rank blockwise/local learning methods of the literature according to their biological plausibility (from left to right), and indicate their demonstrated ability to scale to large-scale datasets (e.g., ImageNet) by the intensity of the blue rectangle below each model family. Our method is situated at a unique trade-off between biological plausibility and performance on large-scale datasets, by being on par in performance with intertwined blockwise training (Xiong et al., 2020) and supervised broadcasted learning (Nøkland, 2016; Clark et al., 2021; Belilovsky et al., 2019) while being more biologically plausible than these alternatives. Methods represented with magenta color used a similar methodology as ours (Halvagal & Zenke, 2022; Löwe et al., 2019), but have only been demonstrated to work on small datasets.

video prediction, using a stack of VAEs trained sequentially in a greedy fashion (Wu et al., 2021). From a neuroscientific standpoint, it is interesting to explore the viability of alternative learning rules to backpropagation, as it is debated whether the brain performs backpropagation (mostly considered implausible) (Lillicrap et al., 2020), approximations of backpropagation (Lillicrap et al., 2016), or relies instead on local learning rules (Halvagal & Zenke, 2022; Clark et al., 2021; Illing et al., 2021). Finally, local learning rules could unlock the possibility for adaptive computations, as each part of the network is trained to solve a subtask in isolation, naturally tuning different parts to solve different tasks (Yin et al., 2022; Baldock et al., 2021), offering interesting energy and speed trade-offs depending on the complexity of the input. There is evidence that the brain also uses computational paths that depend on the complexity of the task (e.g., Shepard & Metzler (1971)).

Self-supervised learning has proven very successful as an approach to pretrain deep networks. A simple linear classifier trained on top of the last layer can yield an accuracy close to the best-supervised learning methods (Chen et al., 2020a;b; He et al., 2020; Chen et al., 2020c; Caron et al., 2021a;b; Zbontar et al., 2021; Bardes et al., 2022; Dwibedi et al., 2021). Self-supervised learning objectives typically have two terms that reflect the desired properties of the learned representations: the first term ensures invariance to distortions that do not affect the label of an image, and the second term ensures that the representation is informative about its input (Zbontar et al., 2021). Such loss functions may better reflect the goal of intermediate layers compared to supervised learning rules which only try to extract labels, without any consideration of preserving input information for the next layers. In this paper, we revisit the possibility of using local learning rules as a replacement for backpropagation on a large scale dataset. As a natural transition towards purely local learning rules, we resort to blockwise training where we limit the length of the backpropagation path to single blocks of the network[1]. We apply Barlow Twins, a recently proposed self-supervised learning loss, locally at different blocks of a ResNet-50 trained on ImageNet and make the following contributions:

---

[1]We consider all layers with the same spatial resolution to belong to the same block.

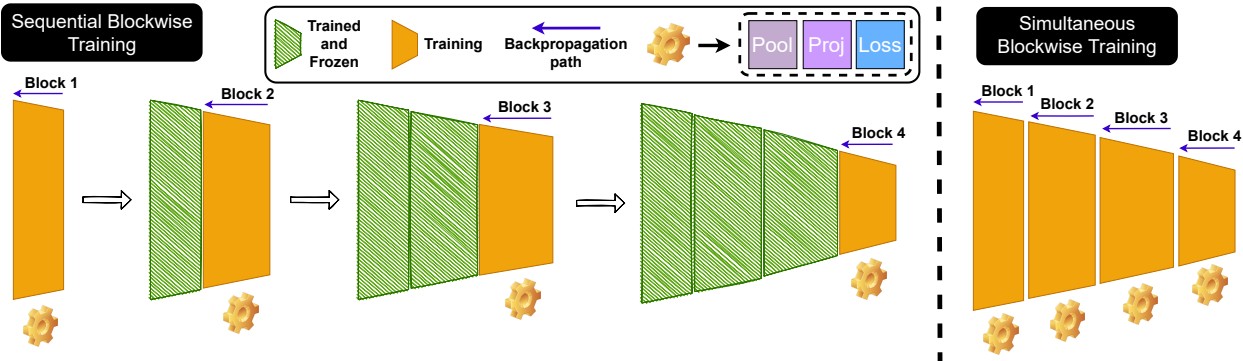

Figure 2: **Overview of our blockwise local learning approach**. Each of the 4 blocks of layers of a ResNet-50 is trained independently using a self-supervised learning rule and a local backpropagation path. We apply this procedure in two main settings: (left) sequential training, where each block, starting from the first block, is independently trained and frozen before the next block is trained; (right) simultaneous blockwise training, where all the blocks are trained simultaneously using a 'stop-grad' operation to limit the backpropagation paths to a single block. The yellow gear symbol refers to the combination of the pooling layer, projection head, and loss function used in self-supervised learning.

- We show that a ResNet-50 trained with our blockwise self-supervised learning method can reach performance on par with the same network trained end-to-end with backpropagation, as long as the network is not broken into too many blocks.

- We find that training blocks simultaneously (as opposed to sequentially) is essential to achieving this level of performance, suggesting that important learning interactions take place between blocks through the feedforward path during training.

- We find that methods that increase downstream performance of lower blocks on the classification task, such as *supervised* blockwise training, tend to *decrease* performance of the overall network, suggesting that building invariant representations prematurely in early blocks is harmful for learning higher-level features in latter blocks.

- We compare different spatial and feature pooling strategies for the intermediate block outputs (which are fed to the intermediate loss functions via projector networks), and find that expanding the feature-dimensionality of the block's output is key to the successful training of the network.

- We evaluate a variety of strategies to customize the training procedure at the different blocks (e.g., tuning the trade-off parameter of the objective function as a function of the block, feeding different image distortions to different blocks, routing samples to different blocks depending on their difficulty), and find that none of these approaches added a substantial gain to performance. However, despite negative initial results, we consider these attempts to be promising avenues to be explored further in future work. We exhaustively describe these negative results in Appendix A.

## 2   Related Work

This section provides a comparison between the existing local learning paradigms in the literature and our method as summarized in Fig. 1. We compare these methods from two perspectives: biological plausibility and demonstrated performance on large-scale datasets, such as ImageNet (Deng et al., 2009).

End-to-end backpropagation (He et al., 2016) is the best-performing paradigm on large-scale datasets, but it is also considered the least biologically plausible (although not impossible, see (Lillicrap et al., 2020)), because it requires a long-range backpropagation path with symmetric weights to the feed-forward path.

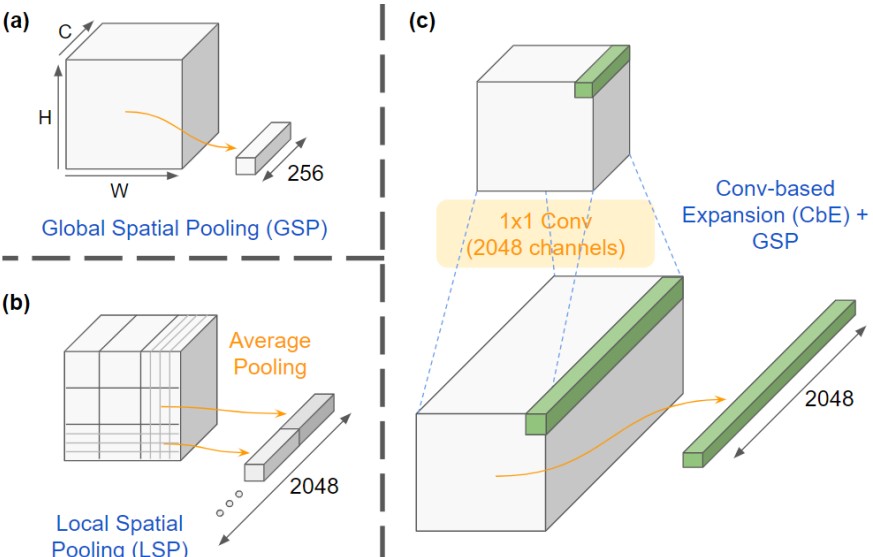

Figure 3: **Overview of the pooling strategies employed in this work**. The first one (a) is the simple Global Spatial Pooling (**GSP**) which simply computes a global average activation over the entire feature volume. The second one (b) is the Local Spatial Pooling (**LSP**) where we divide the feature volume into small spatial bins, compute the averages in these small bins, and concatenate these outputs to compute the final feature vector of size 2048. The third one (c) is a (1x1) Conv-based Expansion (**CbE**), followed by global spatial pooling. This last pooling strategy provides the best performance for our method.

Feedback alignment (Lillicrap et al., 2016) alleviates the symmetry constraint by making the backpropagation path to be random and independent from the feed-forward path. However, this method is not explicitly focused on reducing the length of the backpropagation path.

Nøkland (2016); Clark et al. (2021); Belilovsky et al. (2019); Hinton (2022) are four different methods that avoid long-range backpropagation paths by directly broadcasting the labels to all layers. Therefore, these methods are more biologically plausible than backpropagation. However, all these methods are supervised and thus rely on a large number of labeled inputs, which itself is not biologically plausible.

LoCo (Xiong et al., 2020) applies backpropagation separately to different blocks of a ResNet-50. This method does not rely on an abundance of labels as it uses a self-supervised objective, and performs well on large-scale datasets (69.5% top-1 accuracy on ImageNet with a linear probe). However, LoCo introduces a coupling between subsequent blocks, by applying backpropagation to intertwined pairs of successive blocks. Therefore, this method still conceptually relies on a full backpropagation path.

Our method and the closely related methods (Halvagal & Zenke, 2022; Löwe et al., 2019) avoid any coupling between subsequent blocks by training each block independently, each using a variant of self-supervised learning, namely SimCLR (Chen et al., 2020a), Barlow Twins (Zbontar et al., 2021) and VicReg (Bardes et al., 2022). The main difference between our method and (Halvagal & Zenke, 2022; Löwe et al., 2019) is that we demonstrate the effectiveness of our method on a large-scale dataset i.e. ImageNet. In terms of biological plausibility, our method still relies on backpropagation within blocks and also relies on interactions between neurons within layers due to the exact form of the loss function. These remaining dependencies make our method only partially local, and thus not fully biologically plausible.

Finally, methods such as (Illing et al., 2021; Tang et al., 2022; Journé et al., 2022) avoid all learning dependencies within and across layers and propose truly local learning rules, but these methods have only partially been demonstrated to perform well on large-scale datasets to date.

## 3    Methods

We use a ResNet-50 network and adapt the Barlow Twins (Zbontar et al., 2021) codebase[2] to a blockwise training paradigm.

### 3.1    Blockwise Training

We divide the ResNet-50 network into 4 blocks and limit the length of the backpropagation path to each of these blocks. The ResNet-50 architecture is comprised of 5 blocks of different feature spatial resolutions, followed by a global average pooling operation and a linear layer for final classification. The first block of spatial resolution is only comprised of a single stride-2 layer. For simplicity, we integrate this layer within the first training block, obtaining 4 training blocks. We train each of the 4 blocks separately with a self-supervised learning objective, using stop-gradient to ensure that the loss function from one block does not affect the learned parameters of other blocks (see Fig. 2). We consider two different settings: (i) sequential blockwise training where we train the blocks in a purely sequential way, i.e. one after the other, and (ii) simultaneous blockwise training where we train all the blocks simultaneously. We always report top-1 accuracy on ImageNet by training a linear classifier on top of the frozen representations at the output of the different blocks.

### 3.2    The Barlow Twins loss

Most of the experiments in the paper rely on the Barlow Twins objective function (Zbontar et al., 2021), which we briefly describe here. The input image is first passed through the deep network and then passed to a two-layer projector network in order to obtain the embeddings to be fed to the self-supervised learning loss[3]. We compute these embeddings for two different image views (each image being transformed by a distinct set of augmentations) $z_i^A = \Phi(\Psi(x_i^A))$ where $\Psi$ represents the ResNet-50 encoder, and $\Phi$ represents the projection head. We then define a cross-correlation matrix over these embeddings:

$$\mathcal{C}_{ij} \triangleq \frac{\sum_b z_{b,i}^A z_{b,j}^B}{\sqrt{\sum_b \left(z_{b,i}^A\right)^2} \sqrt{\sum_b \left(z_{b,j}^B\right)^2}} \tag{1}$$

where $b$ indexes batch samples and $i, j$ index the vector dimension of the networks' outputs.

The Barlow Twins objective function attempts to maximize the on-diagonal terms of the cross-correlation matrix and minimize the off-diagonal terms using the following loss function:

$$\mathcal{L}_{\mathcal{BT}} \triangleq \underbrace{\sum_i \left(1 - \mathcal{C}_{ii}\right)^2}_{\text{invariance term}} + \lambda \underbrace{\sum_i \sum_{j \neq i} \mathcal{C}_{ij}^2}_{\text{redundancy reduction term}} \tag{2}$$

where the *invariance term*, by ensuring that on-diagonal elements are close to 1, ensures that the extracted features are invariant to the applied augmentations, while the *redundancy reduction term*, by ensuring that off-diagonal elements are close to 0, ensures that the different units (i.e., neurons) forming the final image representation capture distinct attributes of the input image. The redundancy reduction term thus encourages the representation to be richly informative about the input.

Despite most of our investigation being restricted to Barlow Twins, our findings generalize to other self-supervised learning (SSL) losses such as VicReg (Bardes et al., 2022) or SimCLR (Chen et al., 2020a), as shown in the result section. We implemented SimCLR loss function (Chen et al., 2020a) within the

---

[2]`https://github.com/facebookresearch/barlowtwins`
[3]Using an additional projection head instead of learning directly on the backbone features has been found to be more effective and useful (Chen et al., 2020a).

Barlow Twins codebase and directly adapted the official VicReg implementation[4] for our experiments with VicReg (Bardes et al., 2022).

### 3.3 Pooling Schemes

The original Barlow Twins projector receives an input of dimension 2048. However, the initial blocks of a ResNet-50 have a smaller number of channels. For example, the first block contains only 256 channels instead of 2048. Therefore, we explored different possible ways of pooling these features spatially in order to construct the feature vector that is fed to the projector.

Fig. 3 illustrates the different pooling strategies evaluated in this study, comprised of (1) the Global Spatial Pooling **(GSP)** strategy, which consists in applying a simple average spatial pooling operation to the block's output layer, such that the projector's input dimensionality equals the channel dimensionality of the block's output layer (e.g., 256 for the first block), (2) a Local Spatial Pooling **(LSP)** strategy, where we divide the block's output layer into spatial bins, spatially pool within those bins locally, and then concatenate the outputs of these pooling operations to ensure that the input dimensionality to the projector is 2048, (3) a Conv-based Expansion **(CbE)** pooling strategy, which first expands the number of features to 2048 using the defined convolutional filter size (defaults to $1 \times 1$ convolution), and then performs global spatial pooling. Finally, we evaluate alternatives to the last variant, where we keep an expansion layer, but instead of global spatial pooling, we apply L2 or square-root pooling.

### 3.4 Training

We follow closely the original Barlow Twins training procedure (Zbontar et al., 2021). All our results are for a ResNet-50 trained for 300 epochs, using the LARS optimizer with a batch size of 2048. We use a cosine learning rate decay with an initial learning rate of 0.2. The projector is a three-layered MLP with 8192 hidden units and 8192 output units. In the supervised training case, we use the same training procedure as for the self-supervised case, except that we restrict the output layer of the projector to have 1000 output units, corresponding to the 1000 classes of ImageNet, and apply a simple cross-entropy loss to the output of the projector for classification. For consistency, we also use Barlow Twins' data augmentations in the supervised case. We provide the PyTorch pseudocode for blockwise training of models in Appendix E.

## 4 Matching Backpropagation Performance with Blockwise Training

**A blockwise training procedure with a self-supervised objective leads to accuracies comparable with an end-to-end trained network.** We trained simultaneously the 4 blocks (see Section 3 for details) of a ResNet-50 on ImageNet, using the Barlow Twins loss function at the end of each block. Before being sent to the projector, the output of each block is fed to an expansion-based pooling layer (see Section 3.3 for a description of these pooling layers). We also add noise to the feature maps of each module with a standard deviation of $\sigma = 0.25$. Remarkably, we find that despite each block being trained independently (aside from the forward propagation interactions), our model achieves competitive performance against an end-to-end trained model: we see a gap of only $\sim 1.1\%$ in performance as highlighted in Fig. 4. This is a non-trivial result, as one usually expects a large drop in performance when interactions between blocks are restricted to only forward dynamics on large-scale datasets (Illing et al., 2021; Xiong et al., 2020). Results on the smaller CIFAR-10 dataset are consistent with these results, as presented in Appendix D.

**Extending to other SSL methods: SimCLR / VicReg.** We tested our blockwise paradigm using alternative self-supervised learning rules i.e. VicReg and SimCLR. We visualize these results in Fig. 5. The results we obtain on both VicReg and SimCLR are consistent with our results on Barlow Twins: we only observe a slight loss in accuracy as we move from end-to-end trained networks to expansion-based pooling with simultaneous blockwise training, indicating that our blockwise learning paradigm is robust to the specific choice of the self-supervised learning rule used.

---

[4]`https://github.com/facebookresearch/vicreg`

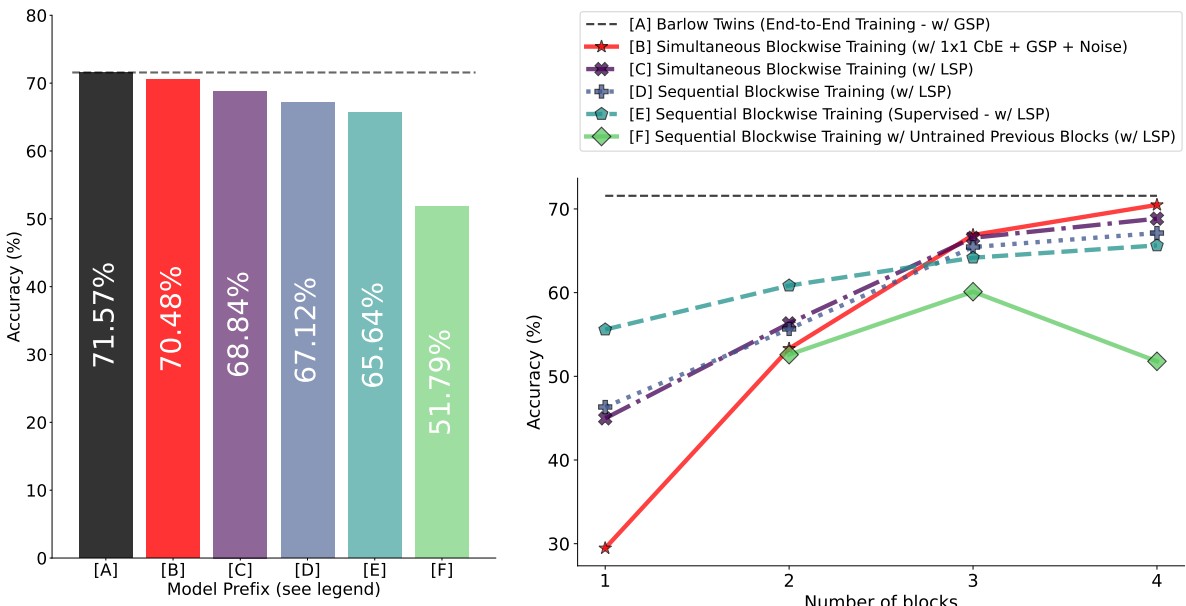

Figure 4: **Overview of our key results.** (left) Top-1 accuracy of a linear probe trained on top of our pretrained network on ImageNet for different pretraining procedures. Our best blockwise-trained model [B] is almost on par with full backpropagation [A] (only 1.1% performance gap). (right) Accuracy on ImageNet as a function of the depth of the network. Since our blockwise models are trained for each block in isolation, we can include a different number of blocks when computing the linear probe accuracy. We visualize the accuracy as we include more blocks into the network.

**Robustness against natural image degradation (ImageNet-C).** We evaluate the robustness of our best model (simultaneous blockwise training w/ 1x1 CbE + GSP + Noise) against natural image degradation using the ImageNet-C benchmark (Hendrycks & Dietterich, 2019). The results are presented in Appendix C. In summary, our results show that our blockwise training protocol degrades robustness in comparison to the end-to-end trained model. This reduction in robustness can be in part attributed to the lower top-1 accuracy of our blockwise trained model.

## 5 Understanding the Contribution of Different Components

We now analyze and discuss the contribution of the different components of our method to build an in-depth understanding of its success. For simplicity, we avoid noise addition to the layers unless we analyze its impact explicitly. Furthermore, we fix the block output pooling strategy to be local spatial pooling unless specified otherwise. We list some further ablations in Appendix B.

**Importance of using a self-supervised learning rule as opposed to a supervised learning rule.** We first compare our blockwise model trained with an SSL loss against a supervised blockwise training procedure, where we train each of these blocks using supervised learning. For a fair comparison, we include the full projection head in the supervised case, but instead of applying the Barlow Twins loss, we project to the 1000 classes of ImageNet and use the class labels for supervising the model at each of these intermediate blocks. We observe that the performance on the classification task of the initial blocks is significantly higher in the supervised learning case than in the SSL case, making the initial block very task-specific, while the latter blocks see a reverse trend. This observation points to an advantage of SSL over supervised learning for successful blockwise learning: whereas SSL rules preserve information about the input in initial blocks for further processing by later blocks, supervised learning objectives already discard information at the initial blocks which could have been used by blocks stacked on top of them.

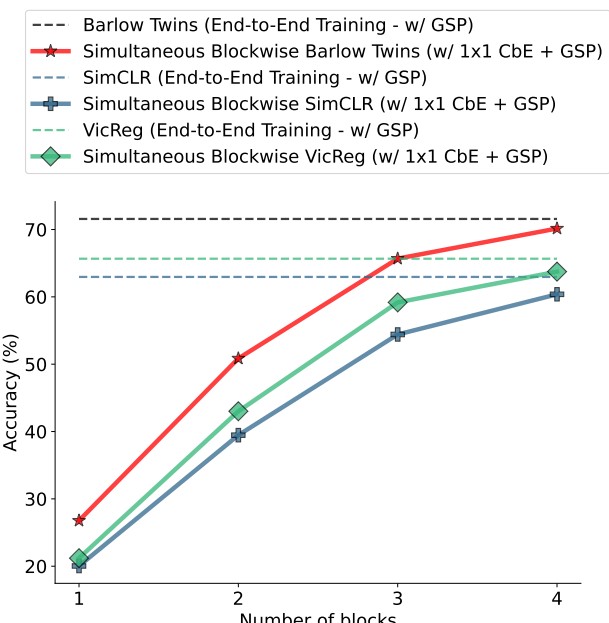

Figure 5: Impact of the SSL objective function used. Our approach (simultaneous blockwise training with conv-based expansion pooling of the block outputs) almost matches end-to-end training performance for all SSL methods tested.

**Importance of simultaneously training all the blocks.** We compare the scheme where we simultaneously optimize the blocks to a sequential optimization scheme. In the sequential optimization scheme, we train each of the blocks in a sequential manner where we freeze the parameters of the previous block (including batch-norm statistics i.e. mean and standard deviation) before starting to train the next block. We see from Fig. 4 that sequential training results in a modest drop in performance with the model accuracy dropping from 68.84% (simultaneous) to 67.12% (sequential)[5]. This decrease in performance can be attributed to two possible reasons: (1) during simultaneous training, the learning trajectory in latter blocks is affected by the learning trajectory co-occurring in earlier blocks and this results in an overall better learning trajectory for the model, or (2) the batch-norm statistics could be interacting in subtle ways during simultaneous training which is beneficial. We tested the second hypothesis by adapting the batch-norm means and variances in early blocks during the training of all the subsequent blocks i.e. by keeping the batch-norm parameters always in training mode. This change makes a negligible impact on the end results, indicating that it is the optimization trajectory taken by these blocks together during training that is beneficial for downstream performance.

**Importance of our custom global average pooling strategy.** Fig. 6 highlights the results that we obtain when comparing the different pooling strategies presented in Fig. 3. We see that conv-based expansion w/ global spatial pooling yields the best final accuracy. Local spatial pooling and global spatial pooling strategies obtain similar performances at the final block, which is slightly inferior to the performance obtained by conv-based expansion w/ global spatial pooling. Conv-based expansion w/ L2-pooling significantly improves the performance of the initial blocks, but deteriorates the performance of the final blocks accordingly (this trend is similar to the supervised blockwise training case). This could be attributed to the fact that L2-pooling normalizes the feature maps using the L2 norm, which overly emphasizes the highest activations. Square-root pooling attempts the opposite, where it tries to squeeze the large values while amplifying small values. However, the overall trend for Square-root pooling remains similar to the

---

[5]Note that the training procedure for both sequential blockwise and simultaneous blockwise training is equivalent when only the first block is trained.

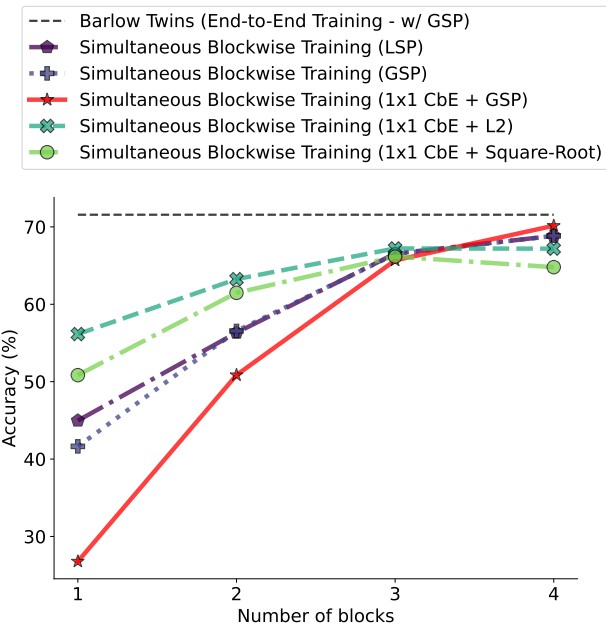

Figure 6: Comparison of different pooling strategies. Alternative pooling strategies show better performance on earlier blocks, but worse performance at the final block.

trend observed for L2-pooling despite being the opposite, with a significant degradation in performance at the final block in contrast to conv-based expansion w/ global spatial pooling.

**Impact of filter size on conv-based expansion pooling.** Our conv-based expansion pooling strategy uses a filter size of $1 \times 1$. We explore the impact of the size of the expansion pooling layer in Fig. 12. We show that increasing the filter size of the expansion pooling layer significantly harms the performance of the model. A possible explanation is that aggregating information over larger neighborhoods (i.e. higher effective receptive field) increases the level of supervision provided at the initial layers, harming the performance at the final blocks.

**Comparison against a trivial baseline of untrained previous blocks.** We next exclude the possibility that the overall accuracy of our blockwise model could be trivially matched by a network of equivalent depth with untrained previous blocks (such random feature models have been used in the past with some success when only training the final layers of the model (Huang et al., 2004)). In order to test this, we compare the accuracy of our model against a model where the previous blocks are untrained, initialized randomly, and frozen. We only train the top block in this case while keeping the (untrained) blocks below fixed. We find that the gap in accuracy between our layer-wise model and the random feature model widens as we stack more blocks without training the initial blocks, as visualized in Fig. 4.

**Impact of the first block on performance.** Correctly training the first block can be supposed to be particularly critical, as all subsequent blocks depend on it. We indeed demonstrate the sensitivity of the model to the exact training procedure of this first block in Fig. 7: by training the first block in an end-to-end setting, and then plugging it into the blockwise training protocol, we largely bridge the gap with end-to-end Barlow Twins (71.57% for the end-to-end variant vs. 70.07% for the variant where the first block is trained end-to-end, while the rest of the blocks are trained in a sequential manner vs. 67.12% for the baseline model trained sequentially). Moreover, if we combine the first two blocks (B1 and B2), and then train them using the same simultaneous blockwise training protocol, this achieves performance on par with using an end-to-end trained first block. This highlights that the main challenge of our blockwise training procedure resides in finding a way to correctly train the first block. We explore different ways to customize the learning process

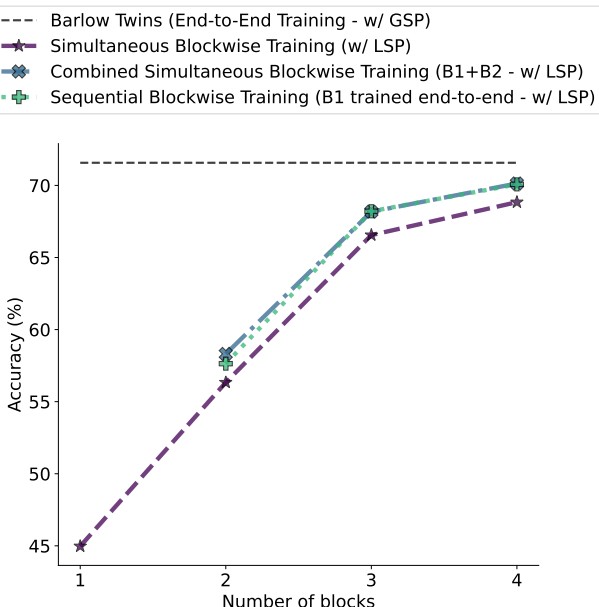

Figure 7: Evaluating the impact of (1) training the first block separately using an end-to-end training procedure, and of (2) merging the first and second blocks during training. Both modifications greatly reduce the performance gap with end-to-end training, indicating that finding a correct training procedure for the first block would be critical to completely match end-to-end training performance.

for the first block in order to improve the performance of the whole network in Appendix A. However, none of these strategies produced the expected gains in performance.

**Impact of noise addition.** Self-supervision in the early blocks may make the blocks too task-specific, reducing overall performance. Therefore, we experiment with the use of noise injected in the activations during training. We pick a particular $\sigma$ and add Gaussian noise with zero mean and the chosen standard deviation to the activations of the model before the beginning of every block. Injecting independent Gaussian noise to each of the feature values with $\mu = 0$ and $\sigma = 0.25$ gave a boost in performance of $\sim 0.5\%$. Using $\sigma = 0.5$ results in lower gains due to high levels of noise (70.22% with $\sigma = 0.5$ vs. 70.48% with $\sigma = 0.25$). We further experimented with different schemes for noise injection such as injecting the same noise to all the features at a particular spatial location, and found independent noise for each of the features to work best. We present the results with noise injection in Fig. 4. Without the addition of noise, our expansion pooling strategy achieves an accuracy of 70.15%.

## 6 Discussion & Conclusion

In this study, we showed that by adapting recent self-supervised learning rules to a blockwise training paradigm, we were able to produce a blockwise training procedure that is comparable in performance to end-to-end backpropagation on a large-scale dataset. Although there is no practical advantage of our method at this point, we believe that our findings are promising from a neuroscientific standpoint, as they propose a viable alternative to end-to-end backpropagation for learning at scale. Moreover, such blockwise training procedures combined with adapted neuromorphic hardware could eventually speed up learning and make it more energy-efficient by eliminating the need for information transportation through the usual backpropagation path.

**Limitations** We are using blocks that do not interact through a backpropagation path. However, each of these blocks from ResNet-50 (He et al., 2016) is composed of a large number of layers as well as a two-layer projector network throughout which the backpropagation algorithm is still used. In order to completely

get rid of backpropagation, one would need to find a learning rule functioning at a layerwise granularity. Unfortunately, our experiments in Fig. 7—showing that merging the two first blocks into one increases the final accuracy of the network—indicate that this gap in performance widens as we progressively subdivide the networks into more blocks. Consequently, our current approach would not be successful in a purely local learning paradigm.

Moreover, Barlow Twins' loss function computes a cross-correlation matrix across all features of a given layer, which implicitly adds an interaction term between all the neurons of this particular layer. In a purely local learning setup, interactions should be limited to local neighborhoods. Illing et al. (2021) proposed such a local learning rule, but they have not demonstrated it to scale to large-scale datasets such as ImageNet.

We find that although there is no backpropagation of information from upper blocks to lower blocks, the feed-forward path still conveys some information that couples the training of the blocks and that this coupling is important, as we show that simultaneous training is better than sequential training from 1.72% points on ImageNet. This means that we cannot yet use our approach to reduce the memory footprint of training by training the blocks sequentially. In future work, one could try to emulate the training of previous blocks by adding specific noise to the previously frozen blocks in the feed-forward path to the block being trained, effectively emulating simultaneous training.

Future work could also be directed toward finding ways to adapt the augmentations' difficulty to the layer being trained. However, the parameter space to explore is large, and our attempts at sorting sample difficulty and routing samples with simple distortions to early blocks have not worked so far (see Appendix A for details). Masked Auto-Encoders (MAE) are an interesting recent class of self-supervised methods based on the Vision Transformer architecture (Dosovitskiy et al., 2020) where the model learns via simple masking of tokens/patches (He et al., 2022). We postulate that our blockwise training framework could be particularly well-suited to such a masking scheme, given that one can adapt the level of difficulty for the augmentations fed to different blocks very naturally via the fraction of patches masked.

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

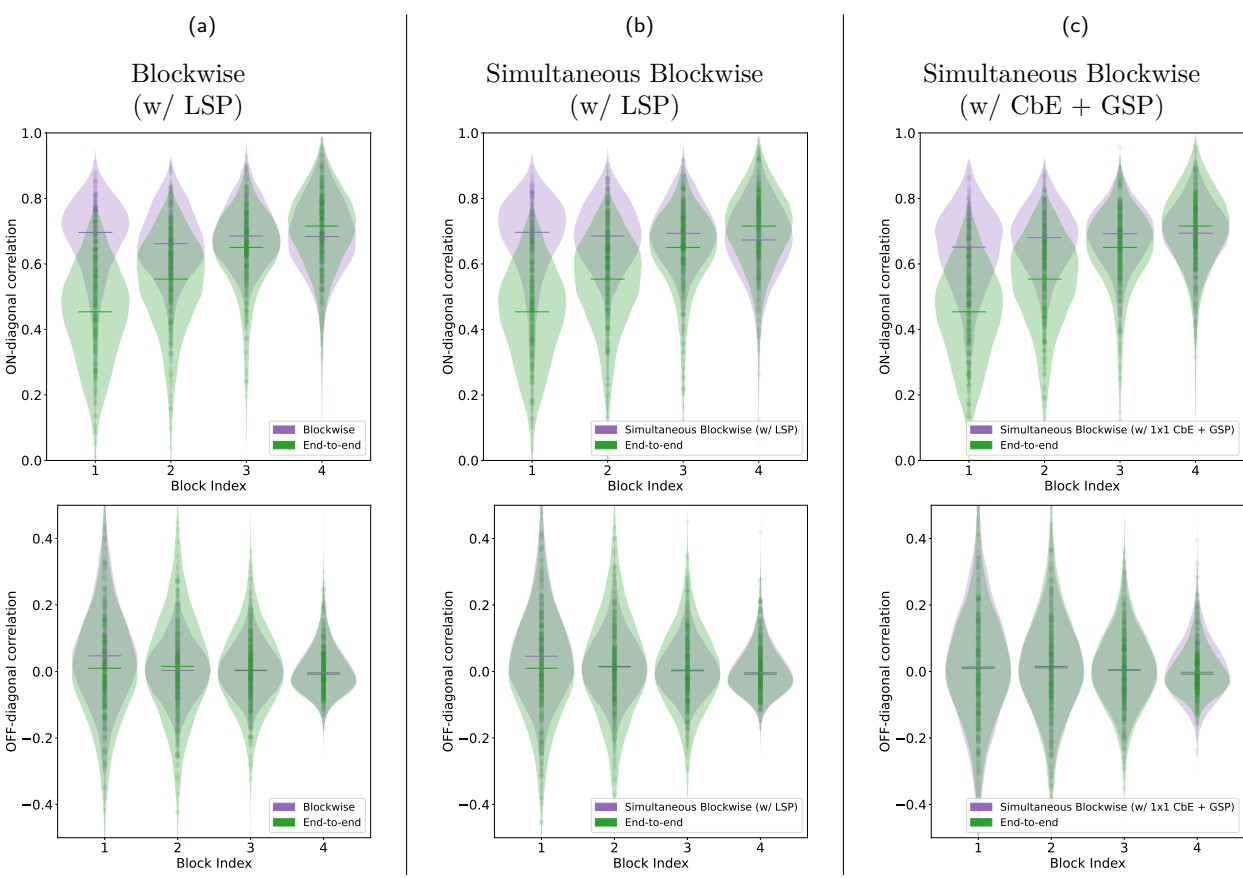

Figure 8: Comparison between ON-diagonal and OFF-diagonal terms of the cross-correlation matrix between features (without projection head or expansion layers and using global average pooling) in comparison with the end-to-end Barlow Twins model. Conv-based pooling improves the alignment on the OFF-diagonal terms, while only improving the alignment of the ON-diagonal terms at the last block.

## A  Customizing the Blockwise Training Process

We tried to customize the training procedure to our blockwise paradigm by progressively increasing the complexity of the task for each block. Although none of these schemes improved the accuracy of the final model, we hope that laying out these schemes and discussing them will aid future research by providing insights on directions worth exploring further.

### A.1  Blockwise Adaptive Learning Strategies

**Analyzing the limitations of the first block.** We visualize the distribution of the on-diagonal and off-diagonal terms from the Barlow Twins loss in the form of a violin plot in Fig. 8. Since the end-to-end trained model has no projection head, we always plot these terms based on the features from the ResNet-50 backbone, without any projection head, and using global average pooling. We see that supervised learning provides a steady increase in the correlation of the on-diagonal terms with an increasing number of blocks, while simultaneously reducing the off-diagonal correlation. However, when looking at blockwise trained models, we see that the initial block is much better at modeling the on-diagonal correlation, while is worse at modeling the on-diagonal correlations in the last block. This again relates back to our observation that providing strong supervision at the initial blocks can worsen the performance of the overall network. We further explore reasons for this mismatch and how they can be catered in the subsequent sections via block combinations as well as tuning of the learning rate or lambda values. It is apparent from Fig. 8 that the first

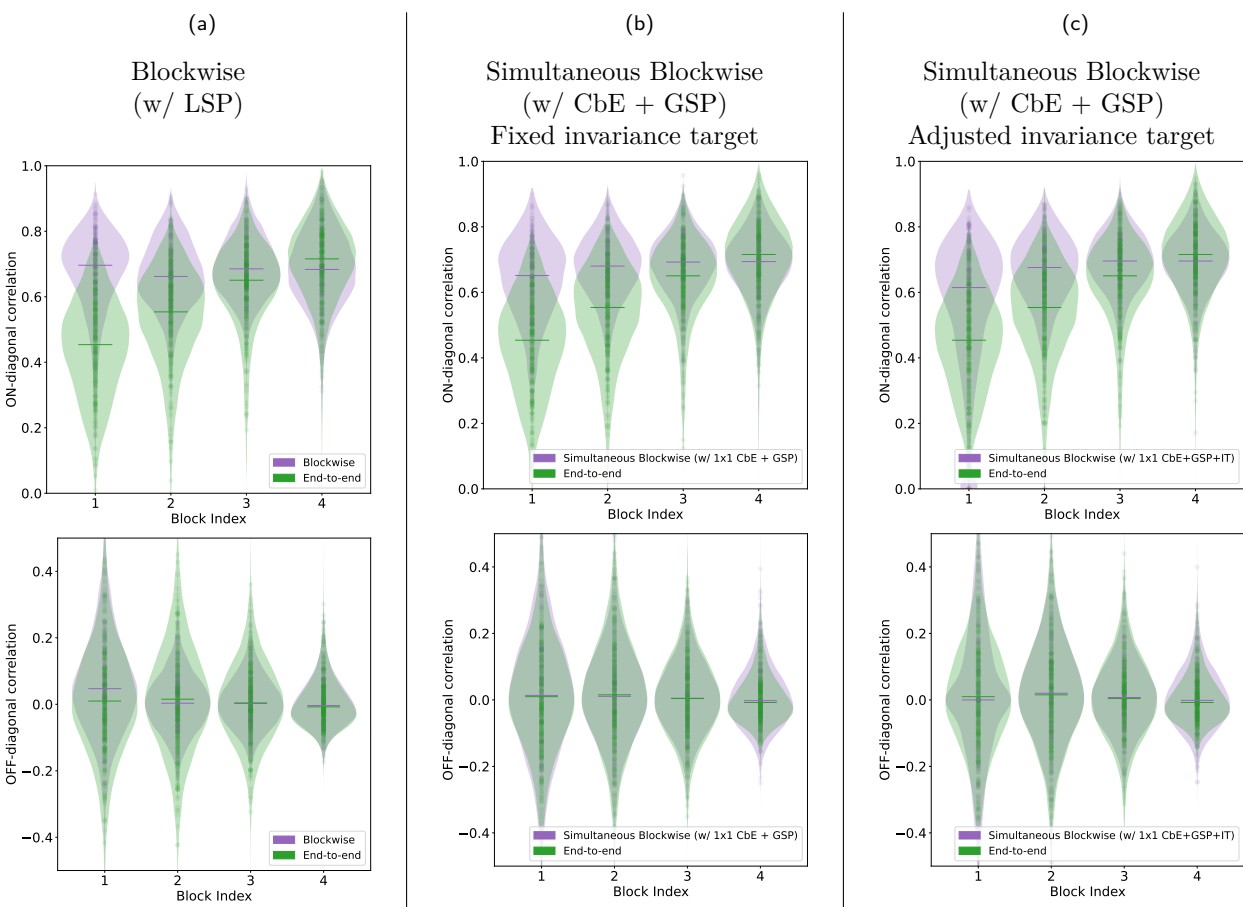

Figure 9: Comparison between ON-diagonal and OFF-diagonal terms of the cross-correlation matrix between features (without projection head or expansion layers and using global average pooling) in comparison with the end-to-end Barlow Twins model and adjusted invariance target model. Conv-based pooling improves the alignment of the OFF-diagonal terms, while only improving the alignment of the ON-diagonal terms at the last block. Variable invariance fails to improve the alignment for the on-diagonal terms in comparison to conv-based pooling while making it worse for the off-diagonal terms. This indicates that the chosen invariance targets are not ideal for this setup, and hence, a different set of targets might be more suitable.

block of the blockwise trained model is significantly different than the end-to-end trained model. Therefore, as a sanity check, we combine the first two blocks of the network. This ensures that the first block is not strictly supervised, and is mainly guided by the performance of the second block. We visualize these results in Fig. 7. We find that this configuration helps in closing the gap between the supervised variant and our block-wise variant where the simultaneous blockwise training achieves 68.84% while the combination of the first two blocks achieves 70.14% top-1 linear probe accuracy. This provides evidence for our initial hypothesis that strong supervision at the initial blocks is significantly detrimental to the downstream performance.

**Matching the Invariance Characteristics of End-to-End Model**  Looking at the difference in distribution between end-to-end and blockwise trained models, it seems likely that minimizing this shift will improve the performance of the model. For this purpose, we evaluate matching the invariance term with that of the end-to-end trained model (redundancy reduction mismatch is already low). One important detail to note is that the difference between the invariance and redundancy reduction terms reported in Fig. 8 is computed based on the application of global spatial pooling to the features of the network without the projection head (as there is no projection head for the end-to-end trained model). Therefore, we use slightly

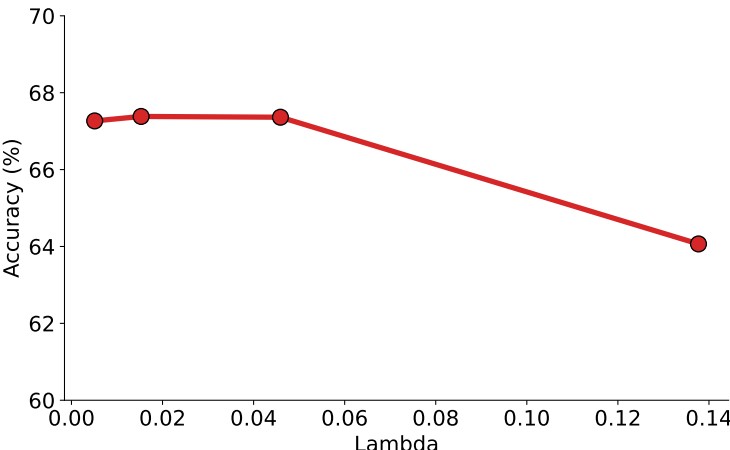

Figure 10: Comparison of different lambda values used to train the model. The default value of $\lambda$ for Barlow Twins is $\lambda = 0.0051$. We see that changes in $\lambda$ values result in a negligible change in the model performance, except when the $\lambda$ value becomes significantly large, where it has a significantly detrimental effect on performance.

higher values for the target invariance, changing it from 1.0 for all the blocks to 0.6, 0.75, 0.9, and 1.0 for block-1, block-2, block-3, and block-4 respectively. However, we found that this fails to improve performance (results instead in an insignificant drop in performance from 70.15% for our model with CbE + GSP to 69.90% with the new invariance targets). We also visualize the feature difference (similar to Fig. 9, but for the model trained with adapted invariance targets). It shows that the chosen target was higher than the one desired. Therefore, a different schedule for targets can potentially outperform the current method in practice.

**Lambda tuning does not help** This experiment was based on blockwise training. In order to reduce the computational burden, we trained the rest of the blocks end-to-end by only adjusting the lambda value for the first block. One recurring theme that we observed throughout our previous experiments was the inferior results when trying to closely supervise the initial layers of the model. We find that supervising these initial layers makes them task-specific, reducing their overall utility. In order to circumvent this issue, we tune the lambda value which provides a trade-off between obtaining invariance to the applied augmentations or minimizing correlations between different features. We assume that enforcing invariance at these initial layers can be premature, however, enforcing diversity in the extracted features by minimizing their correlation can still be beneficial. The Barlow Twins implementation used a very small value of $\lambda = 0.0051$. Therefore, we experimented with larger lambda values. These results are visualized in Fig. 10. We see a negligible impact of changing the lambda value. However, as the value of lambda gets significantly larger, this reduces the performance of the model significantly.

**Adaptive augmentations** Considering that tuning the lambda value made a negligible contribution to the model performance, we tuned the model augmentations such that easier augmentations are applied at the initial blocks and the difficulty of the augmentations increases gradually with higher blocks. This is beneficial as initial blocks can easily learn invariance against color jitters. However, learning invariance against large random crops can be very difficult due to the limited receptive field. Since the dataset is changing, we experiment with this protocol in a strictly sequential setting (where we train the blocks sequentially instead of training them simultaneously). For this purpose, we defined a successive augmentation list, where we only applied color jitter at the first block. We additionally applied small random crops at the second block. For the third and fourth blocks, we applied the full set of Barlow Twins augmentations. We visualize the results from this experiment in Fig. 11. We see that adaptive augmentations result in a significant drop in performance. However, it might be due to some missing components that we were unable to properly tune.

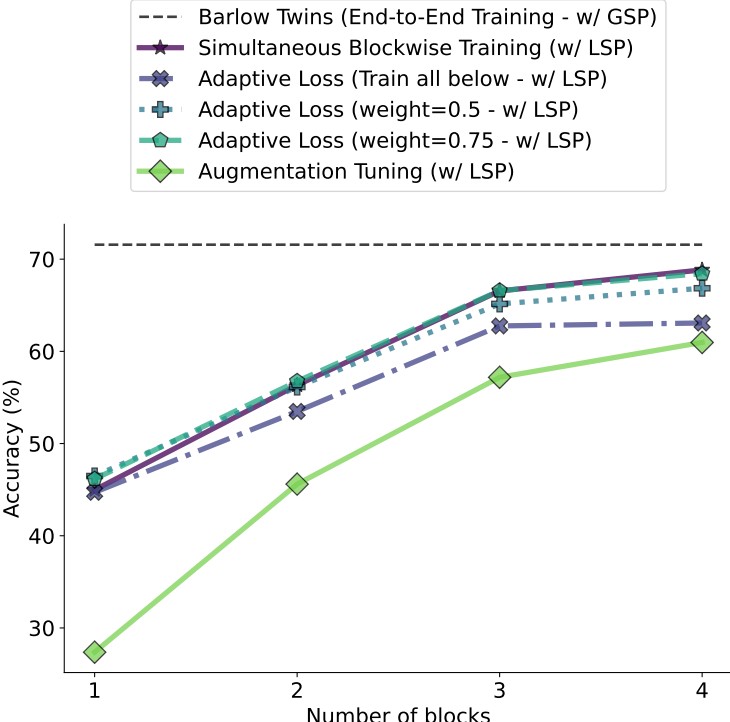

Figure 11: Evaluation of performance with adaptive augmentations or loss functions. Our formulation of adaptive augmentations degrades performance in contrast to regular training where all the blocks are trained using identical augmentations. Similarly, our formulation of the adaptive loss function also degrades performance.

We consider this to be an interesting and open problem for further research and can result in non-trivial gains.

**Example difficulty-based routing** The previous setting with adaptive augmentations can introduce disparity due to differences in the patches seen during the training of the initial blocks vs. training of the later blocks. Therefore, we experimented with a more principled approach of routing the examples to the correct block based on their difficulty by leveraging their training dynamics i.e. routing examples with the smallest loss to the initial blocks while routing examples with the highest loss to the highest blocks. However, it is still important to train all the other blocks aside from the primary block an example is routed towards. For this reason, we tested multiple schemes. The first scheme is [TRAIN ALL BELOW] setting where an example when routed to a particular block, also trains all the blocks below. All the other settings where we specify the weight refers to the setting where we train all the other blocks (not only the blocks below), but with a reduced weight specified using the weight mentioned in the legend. We visualize the results with this routing mechanism in Fig. 11. Similar to the case with adaptive augmentations, we found this routing to reduce the performance of the model significantly. However, we also consider this to be a very interesting avenue to be explored further in future work.

# B  Further Ablations

We performed some trivial ablations by increasing the size of the projector, changing the learning rate of the linear classifier, or even employing group convolutions. However, all these changes had a negligible effect on the model performance.

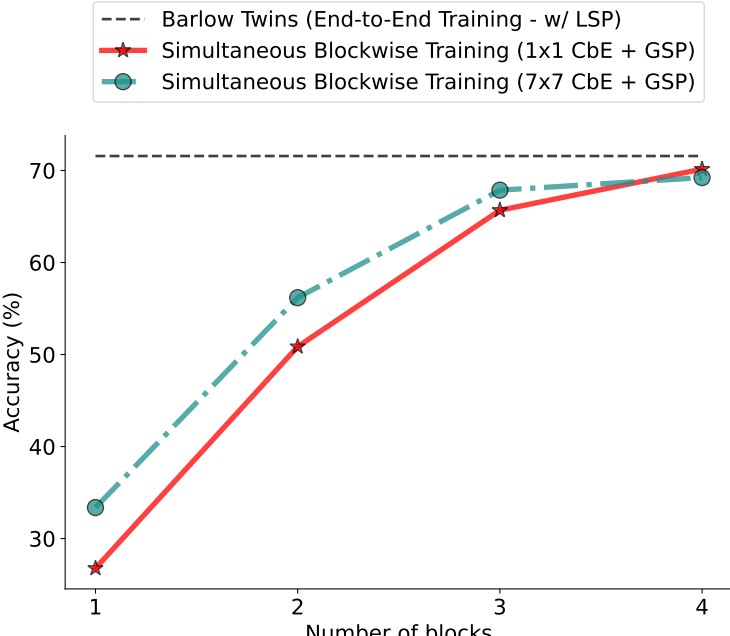

Figure 12: Comparison of performance with changes in filter size for conv-based expansion. Our default setting for conv-based expansion followed by global average pooling uses a filter size of $1 \times 1$. Increasing the size of the filter to $7 \times 7$ improves performance at the initial blocks, but degrades performance towards the end.

**Increasing projector size does not help.** In all the prior experiments, we fixed the projector size to 8192 in accordance with the original Barlow Twins implementation (Zbontar et al., 2021). However, given that local learning is significantly different than end-to-end learning, we experimented with a large projection head. We visualize the impact of large projector size in Fig. 13 where we double the hidden dimensionality of the projection head to 16384. It is interesting to note that increasing the size of the projector increases the performance of the initial blocks but degrades performance at the final block. These results are consistent with the observations made for end-to-end Barlow Twins, where they found a negligible impact of increasing the size of the projection head beyond the limit of 8192 (Zbontar et al., 2021).

**Tuning the learning rate for the linear classifier does not help.** The default learning rate for the linear classification head is set to be 0.3 in the original Barlow Twins evaluation (Zbontar et al., 2021). This was discovered after a grid search over three different values of learning rate. Therefore, in order to identify if a different learning rate is more suitable in our case, we did a small search over the same learning rate settings. These results are summarized in Fig. 14. We found a minimal impact of changing the learning rate, with only deteriorating performance.

**Using group convolutions degrades performance.** Since the expansion-based pooling with $1 \times 1$ convolution was the most impressive in terms of results, we experimented with extending this to group convolution in order to avoid performing this convolution operation over a large spatial grid which is a memory-intensive operation. We visualize these results in Fig. 13. Unfortunately, using group convolutions results in a minor loss in performance in contrast to using the conventional $1 \times 1$ convolution. However, this drop in performance is negligible considering real-world applications.

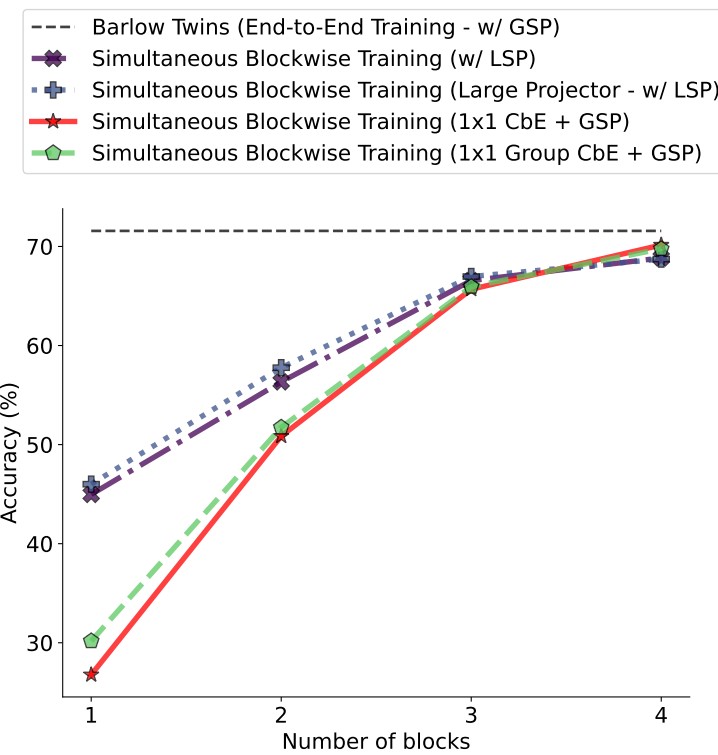

Figure 13: Changing the size of the projection head or using group convolution has a negligible impact on performance.

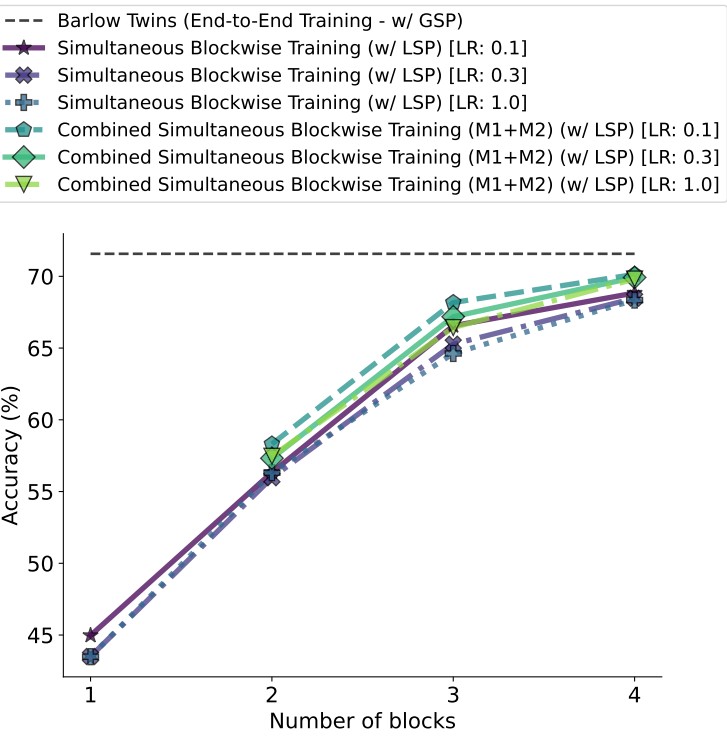

Figure 14: Comparison of different learning rate settings. The default learning rate for Barlow Twins is 0.3. Changes in learning rate have a negligible impact on final model performance.

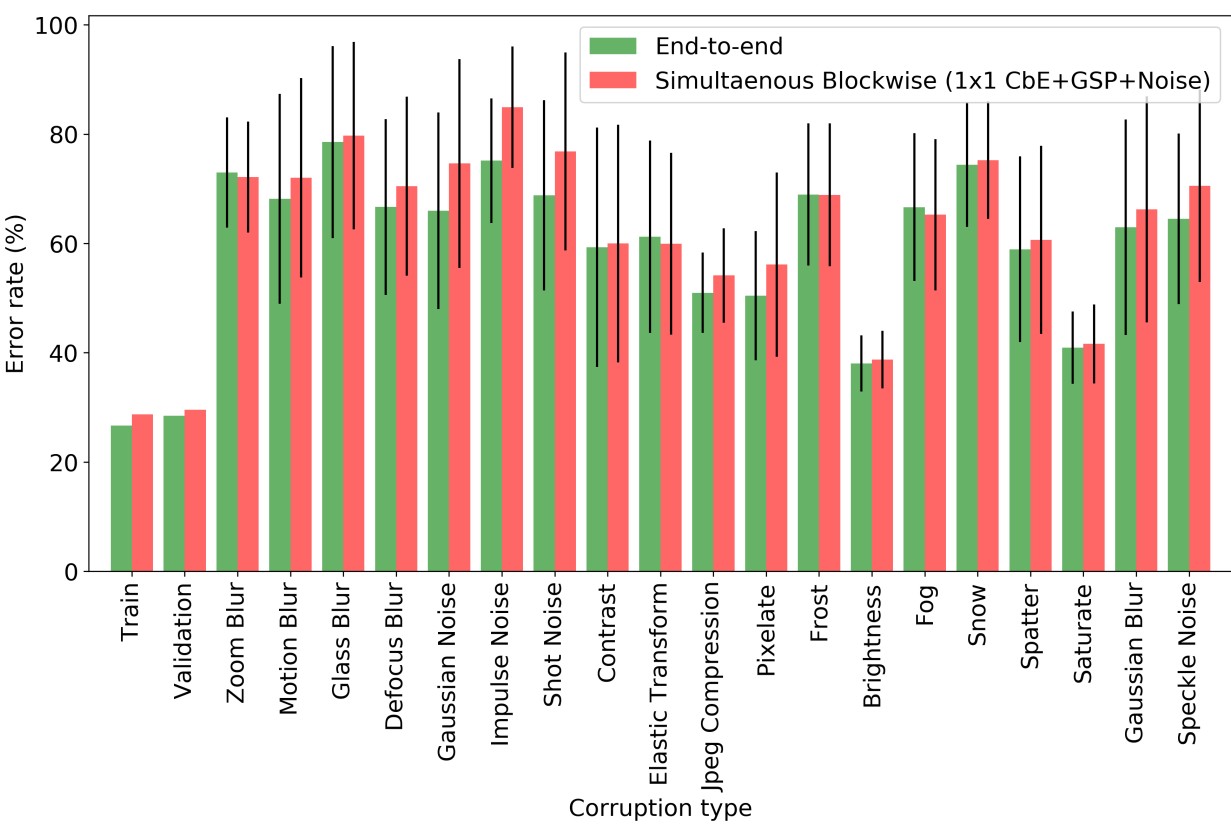

Figure 15: Comparison between error rates of our best model (simultaneous blockwise training w/ 1x1 CbE + GSP + Noise) and the end-to-end trained Barlow Twins model. The figure shows that the blockwise trained model exhibits lower robustness on ImageNet-C distortions as compared to the end-to-end trained model, with an average error rate of 65.68% vs. 62.81% of the end-to-end trained model.

## C Robustness on ImageNet-C

In order to probe our model's robustness to out-of-distribution images, we use ImageNet-C benchmark (Hendrycks & Dietterich, 2019) which was constructed by injecting different kinds of synthetic noise into the ImageNet test set. Each noise type is further segmented into five different levels of noise, ranging from very mild levels of noise to extreme levels of noise. The noise types are selected to cover noise models found in the real world.

We compared the robustness of our best model (simultaneous blockwise training w/ 1x1 CbE + GSP + Noise) and the end-to-end trained Barlow Twins model on ImageNet-C. We report the error rate as well as the standard deviation computed over all five corruption levels for each corruption type. The results are visualized in Fig. 15. The mean error rate (averaged over all corruption types) for the end-to-end trained model is 62.81%, while the mean error rate for our best model is 65.68%. This reduction in robustness can be partly attributed to the lower clean accuracy (71.57% vs. 70.48%). However, these results indicate that blockwise training as operationalized in our case is not particularly useful for robustness against (natural) image degradations.

## D Generalization to CIFAR-10

To showcase the generalizability of our main results to a smaller dataset, we trained the ResNet-50 model on CIFAR-10 from scratch. In order to reuse our augmentation pipeline and model architecture which is

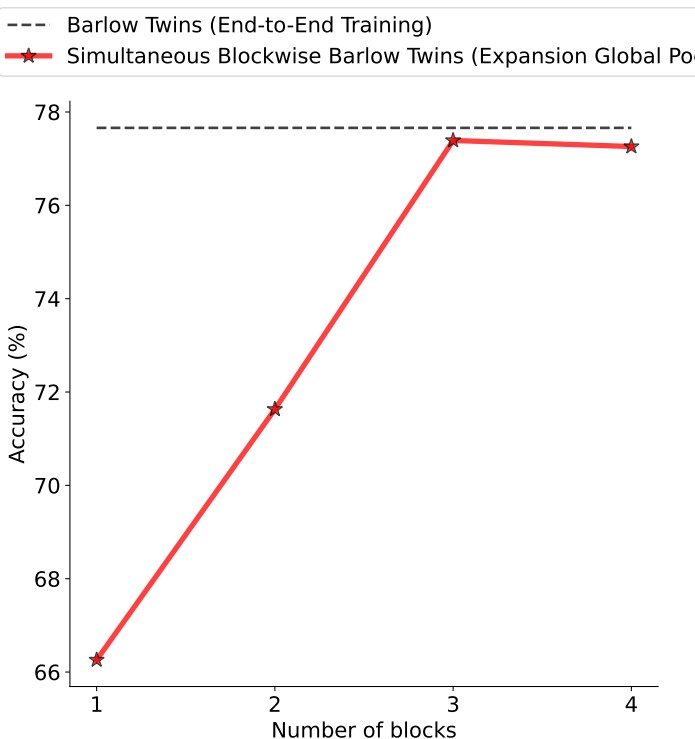

Figure 16: **Demonstration of the generalization of our main results to the smaller CIFAR-10 dataset.** Although CIFAR-10 is relatively small for self-supervised learning to be effective, these results indicate that our main findings generalize to other datasets.

designed assuming $224 \times 224$ images, we upscale the images in CIFAR-10 to $224 \times 224$. The model is trained for 1000 epochs with a smaller batch size of 128 and a learning rate of 0.01, keeping all other settings intact.

Fig. 16 presents a comparison between end-to-end training and our best method on ImageNet i.e., expansion pooling-based sequential blockwise training with a filter size of 1, with no additional noise added to the feature maps. The figure indicates that our results do generalize to other datasets, where the performance of our blockwise approach nearly matches the end-to-end performance. Note that the end-to-end SSL training falls short of the fully-supervised numbers reported on CIFAR-10 for ResNet-50 He et al. (2016) due to the smaller dataset size, use of larger image size as well as the use of ImageNet-optimized augmentations. It is also interesting to note that the performance of the model saturated after the third block, indicating the excess capacity in the model for this smaller dataset.

## E   PyTorch Pseudocode for Blockwise Training

We present the PyTorch pseudocode for our blockwise training scheme in Algorithm 1. Our blockwise training scheme uses PyTorch's 'detach' function (which detaches the tensor from the computational graph) and returns one loss value for each of the 4 blocks of the network. We then backpropagate through each of these 4 loss values individually by calling the 'backward' function.

---

**Algorithm 1:** PyTorch pseudocode for our blockwise training scheme.

---

**Class** *BlockResNet(\*args, \*\*kwargs)*
    **Function** `__init__`(*self*):
        /\* All ResNet initializations                              \*/
    **Function** `forward`(*self, x: Tensor*):
        /\* first layer of the network                    \*/
        $x \leftarrow$ self.maxpool(self.relu(self.bn1(self.conv1(x))))
        output_list $\leftarrow$ []
        $x1 \leftarrow$ self.layer1(x)
        output_list.append(torch.flatten(self.pool_conv_block_1(x1), 1))
        $x2 \leftarrow$ self.layer2(x1.detach())
        output_list.append(torch.flatten(self.pool_conv_block_2(x2), 1))
        $x3 \leftarrow$ self.layer3(x2.detach())
        output_list.append(torch.flatten(self.pool_conv_block_3(x3), 1))
        $x4 \leftarrow$ self.layer4(x3.detach())
        output_list.append(torch.flatten(self.pool_conv_block_4(x4), 1))
        **return** output_list
    **return**
/\* Training code ...                                      \*/
loss_values $\leftarrow$ model(x)
**for** *l in loss_values* **do**
    l.backward()
    // 4 loss values, one for each block
**end**

---

