# OpenReview forum: "Blockwise Self-Supervised Learning at Scale"
_TMLR — Accepted by TMLR_

### Review · Reviewer_iqhT · 2023-10-21

**Summary Of Contributions:**

The paper titled delves into the exploration of alternatives to the conventional backpropagation method, focusing on blockwise learning rules that harness the advancements in self-supervised learning. The authors introduce a blockwise pretraining procedure, where they independently train the four primary blocks of layers of a ResNet-50 using the Barlow Twins' loss function for each block. The results are impressive: a linear probe trained on this blockwise pretrained model achieves a top-1 classification accuracy of 70.48% on ImageNet, which is only 1.1% below the accuracy of a network pretrained end-to-end (71.57% accuracy). This approach is distinct from other methods as it restricts the backpropagation path to individual blocks, and the study demonstrates its effectiveness on large-scale datasets like ImageNet.

The paper also highlights the significance of simultaneous training of blocks, as opposed to sequential training. This is crucial for achieving optimal performance, suggesting that vital learning interactions occur between blocks via the forward propagation path during training. The research indicates that methods enhancing the downstream performance of lower blocks on classification tasks might decrease the overall network's performance. This suggests that building invariant representations too early in the initial blocks could be detrimental to learning higher-level features in subsequent blocks.

**Audience:**

Yes

**Broader Impact Concerns:**

/

**Claims And Evidence:**

Yes

**Requested Changes:**

/

**Strengths And Weaknesses:**

Strengths:

1. The paper presents a novel approach to training deep networks, offering an alternative to the traditional backpropagation method.

2. The results achieved, especially the close performance to end-to-end pretrained networks, are commendable and highlight the potential of the blockwise self-supervised learning approach.

3. The study provides a comprehensive understanding of the impact of different components within the method, contributing to the broader knowledge of local learning rules.

Weaknesses/Questions:

1. How scalable is this approach to even larger networks or more complex architectures? Would the results be consistent across different architectures or datasets beyond ResNet-50 and ImageNet?

2. The biological plausibility of the method is still a topic of debate, as the method relies on backpropagation within blocks and interactions between neurons due to the specific form of the loss function.

3. How does the blockwise approach impact the computational efficiency and training time compared to traditional methods?

---

> ### Author Response · Authors · 2023-11-19
> **Response to reviewer iqhT**
>
> First of all, we would like to thank the reviewer for their precious time in reviewing our manuscript, and for their constructive feedback. We address the interesting questions and issues raised by the reviewer below:
>
> > How scalable is this approach to even larger networks or more complex architectures? Would the results be consistent across different architectures or datasets beyond ResNet-50 and ImageNet?
>
> **Response:** We focused our study on ImageNet and ResNet-50 due to it being the canonical setup for the evaluation of self-supervised learning methods in the past. It would indeed be very interesting to see if our results and insights hold for larger networks and datasets, but we do not have the computational resources to validate this, and leave this for future work. Nevertheless following the reviewer’s suggestion, we have now tested our approach on another dataset, CIFAR-10. We reproduce our main finding that our proposed blockwise training strategy almost matches the performance of an end-to-end trained network:
>
> Blockwise on CIFAR-10 (ResNet-50 w/ expansion pool): 77.26%
>
> End-to-end (ResNet-50): 77.66%
>
> We have now updated the paper with this result (see Appendix D).
>
> > The biological plausibility of the method is still a topic of debate, as the method relies on backpropagation within blocks and interactions between neurons due to the specific form of the loss function.
>
> **Response:** We completely agree that our method is not fully biologically plausible. That’s why our method in Figure 1 is short of the rightmost column, which indicates the highest biological plausibility where only local interactions are allowed. We hope that future work will be able to apply our methodology to the finest granularity where each layer is supervised independently, but this will necessitate finding a way to bridge the widening gap in performance with increasing granularity.
>
> > How does the blockwise approach impact the computational efficiency and training time compared to traditional methods?
>
> **Response:** As we attach additional classification heads to each of the blocks (expansion pooling), our current implementation increases both the memory requirement as well as the training time. For this work, our aim was to understand if such an approach could bridge the performance gap with end-to-end backpropagation. If the effectiveness of such an approach could be established at the finest granularity, future work could focus on designing adaptive computation paradigms, as well as optimized neuromorphic hardware using local learning rules for the efficient training of such networks.

---

### Review · Reviewer_8yqr · 2023-10-23

**Summary Of Contributions:**

This paper proposes a strategy to optimize a Resnet-50 model using local learning rules (i.e. no backprop). The specific strategy here uses a block-wise approach and relies on barlow twins ssl approach. The barlow twins approach operates on the embeddings of a batch distorted in two ways to produce two embeddings for the same batch of data. A cross-correlation matrix is then computed for these two embeddings, and  the loss function has an invariance term, and a redundancy reduction term on the correlation matrix. The resnet-50 model is divided into 4 blocks, and a local version of backprop is applied and propagated only within each block. The primary dataset considered here is ImageNet. The paper shows that this approach scales to imagenet, and remains well performing. Extensive ablation experiments also demonstrate the relative importance of each component of the pipeline here.

**Audience:**

Yes

**Broader Impact Concerns:**

None.

**Claims And Evidence:**

Yes

**Requested Changes:**

Please see the weaknesses listed above. Of those weaknesses, here are the ones I would say they authors should address:
 - The discussion in Section 3.1 needs to be expanded,
-  Explain design decisions,
-  Explain biological plausibility, and
- If possible, test a smaller dataset to show that this method extends to that setting, and explain how the local learning approach here can be applied to say a densenet.

**Strengths And Weaknesses:**

## Strengths
- The paper is well-written and easy to understand.
- Section 5 is particularly important since it helps indicate the relative importance of each of the components of the framework here. For example, the authors assess the impact of noise, the first block, and the SSL training objective. It is particularly interesting to see the effect of a supervised loss vs and self-supervised one. It seems that doing SSL training helps also improve the representations learned by later blocks.
- The paper addresses an important problem of seeking local learning alternatives to backprop.
- The paper demonstrates that they are able to scale the approach to ImageNet models.

## Weaknesses
- **Biological Plausibility**: Figure 1 is interesting, but I am not sure how the authors decided whether an approach is biologically plausible or not. As far as I can tell, evidence for against the biological plausibility of backprop is still unclear, so I am not sure how the authors can get to Figure 1. It seems like the assumption here is that standard backprop is completely not biologically plausible.
- **Design Decisions**: It is unclear to me how the authors picked to partition Resnet-50 into 4 blocks. What is the inspiration for that?
- **Barlow Loss**: The formulation here, as presented, seems excessively tied to the Barlow twins loss. Eventhough their experiments suggest that other SSL losses can also be effective here. If that is the case, what is the point for tailoring this method to the Barlow loss?
- **Is this approach really an alternative to backprop?**: I am not sure we can conclude here that the approach here is an alternative to backprop since you still run a local backprop within each block. The authors mention this in the later parts of the paper.
- **Section 3.1**: I assume this section is meant to be the main meat of the proposed method, but it is too high level for me to really understand what is going on exactly. Can the authors make it clear what the blockwise training does there in detail? Specifically, it would be help to know: given a forward pass, what exactly happens at each block during the local backprop process. I assume, here you backprop within each module as usual (but stopping flow to any parameters outside of that block), but it is not clear if something else happens.
- **Focus on ImageNet and ResNet-50**: The authors tailor most of their findings and analysis to ResNet-50 on image. I'd imagine switching to a different model would be tricky, but it seems like the authors can easily replicate this on a different dataset like celeb-a or another image dataset to test that their findings generalize beyond ImageNet. It will also be interesting for the authors to discuss how this approach can be translated to other architectures, specifically, how should I partition the model? How do you handle the shortcut connections between blocks?

---

> ### Author Response · Authors · 2023-11-19
> **Response to reviewer 8yqr [1/2]**
>
> First of all, we would like to thank the reviewer for their precious time in reviewing our manuscript, and for their constructive feedback. We address the interesting questions and issues raised by the reviewer below:
>
> > Biological Plausibility: Figure 1 is interesting, but I am not sure how the authors decided whether an approach is biologically plausible or not. As far as I can tell, evidence for against the biological plausibility of backprop is still unclear, so I am not sure how the authors can get to Figure 1. It seems like the assumption here is that standard backprop is completely not biologically plausible.
>
> **Response:** Thank you for raising this interesting question. We agree with the reviewer that it is hard to quantify exactly how biologically plausible a particular method is. We consider three different measures of biological plausibility i.e., length of the backpropagation path, dependence within layer, and whether it can be trained in an unsupervised way, criteria which are largely accepted to increase biological plausibility (e.g., Illing et al. 2021). We assume end-to-end backpropagation is the least biologically plausible rather than not biologically plausible at all. Similarly, other methods are also ranked in the same spirit. We consider this summarization to be useful for contextualizing our work, even though this summarization might be in some ways simplistic. We would be grateful to hear from the reviewer if they have suggestions for a better graphical summary of the literature.
>
> *Illing, B., Ventura, J., Bellec, G. and Gerstner, W., 2021. Local plasticity rules can learn deep representations using self-supervised contrastive predictions. Advances in Neural Information Processing Systems, 34, pp.30365-30379.*
>
> > Design Decisions: It is unclear to me how the authors picked to partition Resnet-50 into 4 blocks. What is the inspiration for that?
>
> **Response:** ResNet models are mainly splitted into 5 blocks (each operating at a fixed spatial resolution). Since the first block is just a single layer, we combine the first two blocks into a single block. This splitting of model into spatially fix-sized blocks is a general practice taken from prior work (Xiong et al. 2020).
>
> *Xiong, Y., Ren, M. and Urtasun, R., 2020. Loco: Local contrastive representation learning. Advances in neural information processing systems, 33, pp.11142-11153.*
>
> > Barlow Loss: The formulation here, as presented, seems excessively tied to the Barlow twins loss. Eventhough their experiments suggest that other SSL losses can also be effective here. If that is the case, what is the point for tailoring this method to the Barlow loss?
>
> **Response:** The method is not particularly tied to Barlow Twins. The only reason why we focused on Barlow Twins is the fact that it was the best method at the start of our investigation. Our results do establish that these results are generalizable to other SSL methods.
>
> > Is this approach really an alternative to backprop?: I am not sure we can conclude here that the approach here is an alternative to backprop since you still run a local backprop within each block. The authors mention this in the later parts of the paper.
>
> **Response:** Thanks for raising this. The aim of our investigation was to discover an alternate to end-to-end backpropagation. The reviewer correctly highlighted that we still use backpropagation locally within each block, as mentioned in the paper. We consider the method presented in our paper as a preliminary but necessary step towards discovering a fully local learning rule. Ultimately, if one could find the way to achieve the same performance with a per-layer learning rule, then this would completely remove backprop. Realizing this goal would require further research. We updated section 6 by replacing `an alternative to backpropagation` with `an alternative to end-to-end backpropagation` which covers the reviewer’s concern.

---

> ### Author Response · Authors · 2023-11-19
> **Response to reviewer 8yqr [2/2]**
>
> > Section 3.1: I assume this section is meant to be the main meat of the proposed method, but it is too high level for me to really understand what is going on exactly. Can the authors make it clear what the blockwise training does there in detail? Specifically, it would be help to know: given a forward pass, what exactly happens at each block during the local backprop process. I assume, here you backprop within each module as usual (but stopping flow to any parameters outside of that block), but it is not clear if something else happens.
>
> **Response:** This is correct. Our method is extremely simple i.e., you just add stop-gradient between blocks. In PyTorch, this would mean calling the `.detach()` function after every block of processing. Therefore, backpropagating through each of the different 4 loss functions, one for each block, would only backpropagate up until the end of that block. Considering the reviewer’s clarity concern, we have now added a PyTorch pseudo-code to the draft (Appendix E) in order to make the algorithm clearer (added for reference below).
>
> ```python
> class BlockResNet(*args, **kwargs):
>     def __init__(self):
>         """All ResNet initializations"""
>
>     def forward(self, x: Tensor) -> Tensor:
>         # first layer of the network
>         x = self.maxpool(self.relu(self.bn1(self.conv1(x))))
>
>         output_list = []
>
>         x1 = self.layer1(x)
>         output_list.append(torch.flatten(self.pool_conv_block_1(x1), 1))
>
>         x2 = self.layer2(x1.detach())
>         output_list.append(torch.flatten(self.pool_conv_block_2(x2), 1))
>
>         x3 = self.layer3(x2.detach())
>         output_list.append(torch.flatten(self.pool_conv_block_3(x3), 1))
>
>         x4 = self.layer4(x3.detach())
>         output_list.append(torch.flatten(self.pool_conv_block_4(x4), 1))
>
>         return output_list
>
>
> # Training code …
> loss_values = model(x)
> [l.backward() for l in loss_values]   # 4 loss values, one for each block
> ```
>
> > Focus on ImageNet and ResNet-50: The authors tailor most of their findings and analysis to ResNet-50 on image. I'd imagine switching to a different model would be tricky, but it seems like the authors can easily replicate this on a different dataset like celeb-a or another image dataset to test that their findings generalize beyond ImageNet. It will also be interesting for the authors to discuss how this approach can be translated to other architectures, specifically, how should I partition the model? How do you handle the shortcut connections between blocks?
>
> **Response:** Our partitioning technique is generic in the sense that we divide the network in blocks of layers, and it can be generalized to any architecture. In the presence of skip connections between blocks such as DenseNet, one would need to ensure that both the paths have appropriate stop-gradient mechanisms in place, such that the gradients don’t propagate through the main branch nor the skip connections.
>
> We focused our study on ImageNet and ResNet-50 due to it being the canonical setup for the evaluation of self-supervised learning methods in the past. Following the reviewer’s suggestion, we have now tested our approach on another dataset, CIFAR-10. We reproduce our main finding that our proposed blockwise training strategy almost matches the performance of an end-to-end trained network:
>
> Blockwise on CIFAR-10 (ResNet-50 w/ expansion pool): 77.26%
>
> End-to-end (ResNet-50): 77.66%
>
> We have now updated the paper with this result (see Appendix D)

---

> > ### Comment · Reviewer_8yqr · 2023-12-08
> > **Thanks for the response**
> >
> > I'd like to thank the authors for the response. I particularly appreciated the code since that is exactly what I was curious about. It is also great that the authors put that code block in the appendix. The updated draft addresses all of my concerns. One last issue I'd push the authors on is to discuss the future/open problems in this line of work more carefully. As it stands, one benefit of the proposal is the simplicity of the approach. However, one could argue that it doesn't yet provide enough benefits such that we would switch to the blockwise training. If the benefit is then not engineering specific, then more should be done to discuss alternative benefits. Overall, I think this is interesting work.

---

### Review · Reviewer_84cX · 2023-11-08

**Summary Of Contributions:**

This paper investigates different optimization strategies for ResNet-50-based architectures on ImageNet self-supervised objectives. The strategies investigated focus on 'local' optimization updates, for purported gains in optimization efficiency. Experiments investigate the classification accuracy as a function of the number of ResNet-blocks used, and vary the architecture and optimization. These experiments illustrate that 'simultaneous' blockwise training nearly matches full gradient-based optimization, and outperforms 'sequential' optimization, given certain restrictions on the block architecture (spatial and feature pooling strategies). The paper also presents negative results on other blockwise optimization strategies.

**Audience:**

Yes

**Claims And Evidence:**

Yes

**Requested Changes:**

Address the weaknesses would strengthen the work. In my opinion, the main weaknesses are the absence of evidence that highlights in which cases the proposed method would be preferred, as well as the fact that the results are limited to single dataset and architecture. Despite these weaknesses, the claims are accurately scoped.

**Strengths And Weaknesses:**

# Strengths
- The high-level justification for the method, from the perspective of optimization efficiency, makes sense, which motivates the work.
- The related works discussed appear sufficient
- The experimental protocol description appears fairly reproducible
- The paper is well-structured and easy to read
- The method figures are well-designed and helpful
- The results figures are clear

# Weaknesses
- Abstract does not motivate why the proposed blockwise pretraining procedure is desirable. The only metrics mentioned are ImageNet accuracy, which is a performance degradation. Is the method faster? Cheaper? More amenable to hardware restrictions? By what metrics? I'm hoping to learn more by reading the main text, but a better abstract would be self-contained.
- The experiments do not include a metric that measures the purported benefit(s) of the approach relative to end-to-end training (e.g. energy consumption, memory footprint, etc.). Thus, there is no evidence for why the proposed approach would be preferred to end-to-end training, which achieves superior accuracy. Including measurements of these statistics for each approach would improve the paper.
- Page 2: ".... as long as the network is not broken into too many blocks" where is this result presented? I couldn't find a figure or table where there was significant performance degradation after the addition of "too many" blocks. Ideally the paper would include a reference to the result in the contribution description and make the contribution wording more precise (e.g. "after <X> many blocks").
- The paper organizes optimization approaches along a 'biological plausiblity' spectrum, but there is no self-contained justification for this ordering.
- The experiments are limited to a single base model architecture (ResNet-50)
- The experiments are limited to a single dataset
- The experiments appear to be limited to a single seed per configuration, which does not attempt to isolate possible variance due to initialization and optimization randomness
- Fig. 4 (right). It's not clear what the difference between the Sequential and Simultaneous training methods is when there is only 1 block (Fig. 4 (right), at x=1). Aren't the methods equivalent in this case? The Fig. and the main text should make this clear

---

> ### Author Response · Authors · 2023-11-19
> **Response to reviewer 84cX [1/2]**
>
> First of all, we would like to thank the reviewer for their precious time in reviewing our manuscript, and for their constructive feedback. We address the interesting questions and issues raised by the reviewer below:
>
> > Abstract does not motivate why the proposed blockwise pretraining procedure is desirable. The only metrics mentioned are ImageNet accuracy, which is a performance degradation. Is the method faster? Cheaper? More amenable to hardware restrictions? By what metrics? I'm hoping to learn more by reading the main text, but a better abstract would be self-contained.
>
> **Response:** Thank you for highlighting this. We agree that highlighting the objective as well as the desirable outcome is necessary in the abstract. We have now updated the abstract with the following sentence: “Current state-of-the-art deep networks are all powered by backpropagation. However, long backpropagation paths as found in end-to-end training are biologically implausible, as well as inefficient in terms of energy consumption.”
>
> > The experiments do not include a metric that measures the purported benefit(s) of the approach relative to end-to-end training (e.g. energy consumption, memory footprint, etc.). Thus, there is no evidence for why the proposed approach would be preferred to end-to-end training, which achieves superior accuracy. Including measurements of these statistics for each approach would improve the paper.
>
> **Response:** The main focus of our work is a scientific investigation of whether one can do local learning in the absence of long backpropagation paths (which is considered biologically implausible) without incurring any loss in model utility. We do mention the downstream advantages of reduced energy consumption using neuromorphic hardware or lower memory footprint if such a method can be realized in full glory. However, realizing such advantages is beyond the scope of our work as it requires further research.
>
> > Page 2: ".... as long as the network is not broken into too many blocks" where is this result presented? I couldn't find a figure or table where there was significant performance degradation after the addition of "too many" blocks. Ideally the paper would include a reference to the result in the contribution.
>
> **Response:** We found a drop in performance when we divided the network in 4 blocks compared to 3 blocks (Figure 7). We were unable to investigate performance drops when using even more blocks due to increasing memory demands (each classification head has an expansion pooling layer, which is computationally very expensive). Therefore, these conclusions are an extrapolation from our results. Note that we tie back the results of Figure 7 to the claim of the intro in the Discussion page 10: “Unfortunately, our experiments in Fig. 7—showing that merging the two first blocks into one increases the
> final accuracy of the network—indicate that this gap in performance widens as we progressively subdivide
> the networks into more blocks.”
>
> > The paper organizes optimization approaches along a 'biological plausiblity' spectrum, but there is no self-contained justification for this ordering.
>
> **Response:** Thank you for raising this interesting question. We agree with the reviewer that it is hard to quantify exactly how biologically plausible a particular method is. We consider three different measures of biological plausibility i.e., length of the backpropagation path, dependence within layer, and whether it can be trained in an unsupervised way, criteria which are largely accepted to increase biological plausibility (e.g., Illing et al. 2021). We assume end-to-end backpropagation is the least biologically plausible rather than not biologically plausible at all (notice that the rankings are relative). Similarly, other methods are also ranked in the same spirit. We consider this summarization to be useful for contextualizing our work, even though this summarization might be in some ways simplistic. We would be grateful to hear from the reviewer if they have suggestions for a better graphical summary of the literature.
>
> *Illing, B., Ventura, J., Bellec, G. and Gerstner, W., 2021. Local plasticity rules can learn deep representations using self-supervised contrastive predictions. Advances in Neural Information Processing Systems, 34, pp.30365-30379.*

---

> ### Author Response · Authors · 2023-11-19
> **Response to reviewer 84cX [2/2]**
>
> > The experiments are limited to a single base model architecture (ResNet-50) and a single dataset
>
> **Response:** We focused our study on ImageNet and ResNet-50 due to it being the canonical setup for the evaluation of self-supervised learning methods in the past. Following the reviewer’s suggestion, we have now tested our approach on another dataset, CIFAR-10. We reproduce our main finding that our proposed blockwise training strategy almost matches the performance of an end-to-end trained network:
>
> Blockwise on CIFAR-10 (ResNet-50 w/ expansion pool): 77.26%
>
> End-to-end (ResNet-50): 77.66%
>
> We have now updated the paper with this result (see Appendix D)
>
> > The experiments appear to be limited to a single seed per configuration, which does not attempt to isolate possible variance due to initialization and optimization randomness
>
> **Response:** Although we do report a single run, we ran multiple models at times with different configurations (stopped early instead of training them for the full training budget), and found the variance between runs to be small (< 1% i.e., about the same range as that of general SSL techniques). It would have been ideal to report error bars as the reviewer suggests. However, reporting these error bars would have required multiple runs of an extremely expensive training run.
>
> > Fig. 4 (right). It's not clear what the difference between the Sequential and Simultaneous training methods is when there is only 1 block (Fig. 4 (right), at x=1). Aren't the methods equivalent in this case? The Fig. and the main text should make this clear
>
> **Response:** That’s a great point. They should be approximately equal in this case. We have updated the main text to make this explicit (see footnote on page 8).

---

### Author Response · Authors · 2023-11-19
**Updates to the draft**

We are very thankful to all the reviewers for their useful and constructive comments. Based on these comments, we have made a number of changes to the draft. All changes are marked in RED to make it easier for the reviewers to identify these changes.
- We have added results on the smaller CIFAR-10 dataset, where we show that our main results are also generalizable to this setting. These results are presented in Appendix D. We have also added a reference to the appendix in the main results section.
- We have added PyTorch Pseudocode for blockwise training in Appendix E, in order to make our training setup clear. We have also added a reference to Appendix E in the methods section.
- We have made small updates to the paper based on the reviewers' concerns (again marked in RED).

---

### Decision · Action_Editor_HKuC · 2024-01-23

**Recommendation:** Accept as is

**Comment:**

All of the reviewers were convinced that the experiments support the claims made in the paper: the approach can train vision models in a blockwise fashion without too large of an accuracy degradation. However, reviewers also shared concern about whether this demonstration is significant. Some of the reviewers hoped to see a well defined metric that would benefit from blockwise training, whether in terms of compute efficiency, energy efficiency, or the compute footprint.

Given the TMLR guidelines, I do believe that some individuals in TMLR's audience, especially those working on "biologically plausible" learning or those who are working on alternatives to backprop, will find results in this paper interesting. The evidence is clear that authors can get within ~1% of baseline ResNet-50 trained on ImageNet without using full backprop.

**Audience:**

Researchers interested in biologically plausible learning, alternatives to backprop, memory-efficient learning.

**Claims And Evidence:**

This paper claims to present an approach that can train vision models in a blockwise fashion, without using full backprop. The evidence is mainly the observation that they get within 1.1% of regular ImageNet training, with additional results on CIFAR-10. All of the reviewers seemed convinced that the claims made in this were supported with experiments.